# EPAC1 inhibition protects the heart from doxorubicin-induced toxicity

Marianne Mazevet[1], Anissa Belhadef[1], Maxance Ribeiro[1], Delphine Dayde[1], Anna Llach[1], Marion Laudette[2], Tiphaine Belleville[3], Philippe Mateo[1], Mélanie Gressette[1], Florence Lefebvre[1], Ju Chen[4], Christilla Bachelot-Loza[3], Catherine Rucker-Martin[5,6‡], Frank Lezoualc'h[2], Bertrand Crozatier[1], Jean-Pierre Benitah[1], Marie-Catherine Vozenin[7], Rodolphe Fischmeister[1], Ana-Maria Gomez[1], Christophe Lemaire[1,8†], Eric Morel[1]*†

[1]Université Paris-Saclay, Inserm, UMR-S 1180, Orsay, France; [2]Institut des Maladies Metaboliques et Cardiovasculaires - I2MC, INSERM, Université de Toulouse, Toulouse, France; [3]Innovations Thérapeutiques en Hémostase - UMR-S 1140, INSERM, Faculté de Pharmacie, Université Paris Descartes, Sorbonne Paris Cité, Paris, France; [4]Basic Cardiac Research UCSD School of Medicine La Jolla, San Diego, United States; [5]Faculté de Médecine, Université Paris-Saclay, Le Kremlin Bicêtre, France; [6]Inserm UMR_S 999, Hôpital Marie Lannelongue, Le Plessis Robinson, France; [7]Laboratoire de Radio-Oncologie, CHUV, Lausanne, Switzerland; [8]Université Paris-Saclay, UVSQ, Inserm, Orsay, France

*For correspondence:
eric.morel@universite-paris-saclay.fr

†These authors contributed equally to this work

‡Deceased

Competing interest: The authors declare that no competing interests exist.

**Abstract** Anthracyclines, such as doxorubicin (Dox), are widely used chemotherapeutic agents for the treatment of solid tumors and hematologic malignancies. However, they frequently induce cardiotoxicity leading to dilated cardiomyopathy and heart failure. This study sought to investigate the role of the exchange protein directly activated by cAMP (EPAC) in Dox-induced cardiotoxicity and the potential cardioprotective effects of EPAC inhibition. We show that Dox induces DNA damage and cardiomyocyte cell death with apoptotic features. Dox also led to an increase in both cAMP concentration and EPAC1 activity. The pharmacological inhibition of EPAC1 (with CE3F4) but not EPAC2 alleviated the whole Dox-induced pattern of alterations. When administered *in vivo*, Dox-treated WT mice developed a dilated cardiomyopathy which was totally prevented in EPAC1 knock-out (KO) mice. Moreover, EPAC1 inhibition potentiated Dox-induced cell death in several human cancer cell lines. Thus, EPAC1 inhibition appears as a potential therapeutic strategy to limit Dox-induced cardiomyopathy without interfering with its antitumoral activity.

## Editor's evaluation

The clinical implications of doxorubicin-induced cardiac toxicity are well established. The discovery of novel druggable cardioprotective targets has significant biological and clinical impact.

## Introduction

Despite its frequent use and clinical efficiency, the anticancer anthracycline doxorubicin (Dox) shows strong side effects including cardiotoxicity (*McGowan et al., 2017*; *Tacar et al., 2013*). Dox cardiotoxicity can be acute with immediate cardiac disorders (arrhythmias) or chronic with remodeling cardiomyopathies (dilated cardiomyopathy [DCM], heart failure [HF]) years after treatment (*Lipshultz et al., 2010*).

Mechanistically, Dox, a DNA topoisomerase II inhibitor, induces DNA double strand break by direct DNA interaction (*L'Ecuyer et al., 2006*), oxidative stress (*Cappetta et al., 2017*; *Rochette et al., 2015*), and decreased ATP production (*Tokarska-Schlattner et al., 2006*). This induces apoptosis and necrosis through p53 cascade and ROS production/ATP depletion, respectively (*Liu et al., 2008*) and is proposed to be a primary mechanism of Dox-induced cardiomyopathy (*Zhang et al., 2009*; *Casey et al., 2007*). Dox also induces mitochondrial biogenesis alteration and energy defects (*Goormaghtigh et al., 1990*; *Guo et al., 2014*; *Kavazis et al., 2017*; *Dorn et al., 2015*). DNA damage and HF are prevented in cardiac-specific topoisomerase IIβ (TopIIβ) knock-out (KO) mice, leading to the conclusion that TopIIβ is fundamental in Dox-induced cardiotoxicity (*Zhang et al., 2012*). However, alternative mechanisms have been proposed to explain Dox-induced cardiotoxicity (*Sawicki et al., 2021*). Currently, the only approved medication against anthracyclines cardiotoxicity is dexrazoxane (*Lipshultz et al., 2010*). However, the drug is not devoid of adverse effects, such as potential second malignancies incidence and reduced cancer treatment efficacy (*Swain et al., 1997*; *Reichardt et al., 2018*; *Tebbi et al., 2007*). Thus, there is a need to search for new therapeutics which would provide long-term cardioprotection against Dox-induced cardiotoxicity without compromising its antitumoral efficacy.

The exchange factor EPAC (exchange protein directly activated by cAMP) contributes to the hypertrophic effect of β-adrenergic receptor chronic activation in a cAMP-dependent but PKA-independent manner (*Métrich et al., 2008*; *Morel et al., 2005*). Two EPAC genes are present in vertebrates, EPAC1 and EPAC2 (*Robichaux and Cheng, 2018*). EPAC1, encoded by the *Rapgef3* gene, is the main isoform expressed in human hearts and its expression is increased in HF (*Métrich et al., 2008*). EPAC is a guanine nucleotide exchange factor for the small G-protein Rap (*de Rooij et al., 1998*; *Kawasaki et al., 1998*) and in pathological conditions, promotes cardiac remodeling through pro-hypertrophic signaling pathways, which involve nuclear $Ca^{2+}$/H-Ras/CaMKII/MEF2 and cytosolic $Ca^{2+}$/Rac/H-Ras/calcineurin/NFAT (*Morel et al., 2005*; *de Rooij et al., 1998*; *Pereira et al., 2012*; *Cazorla et al., 2009*; *Oestreich et al., 2009*). Although the role of EPAC in cardiomyocyte apoptosis remains controversial (*Mangmool et al., 2015*; *Okumura et al., 2014*; *Suzuki et al., 2010*), several molecular intermediates in the EPAC1 signalosome (e.g. $Ca^{2+}$ homeostasis, RhoA, Rac) have been shown to be involved in Dox-induced cardiotoxicity (*Sag et al., 2011*; *Riganti et al., 2008*; *Huelsenbeck et al., 2012*). There is also evidence that EPAC inhibition may be cardioprotective in ischemia/reperfusion injury (*Fazal et al., 2017*). Moreover, EPAC was shown to contribute to cardiac hypertrophy and amyloidosis induced by radiotherapy (*Monceau et al., 2014*). Finally, EPAC was shown to play a role in various cancers (*Huk et al., 2018*; *Kumar et al., 2018*) and tumoral processes (*Jansen et al., 2016*; *Almahariq et al., 2015*; *Almahariq et al., 2016*), suggesting EPAC as a potential therapeutic target for cancer treatments (*Kumar et al., 2018*; *Wang et al., 2017*).

All this suggests that EPAC may play a role in Dox-induced cardiotoxicity. Thus, the aim of this study was to explore this hypothesis by evaluating whether inhibition of EPAC1, either pharmacologically (using the specific EPAC1 inhibitor, CE3F4; *Courilleau et al., 2012*) or genetically (EPAC1 KO mice), is cardioprotective against Dox-induced cardiotoxicity.

## Results

### Dox-induced DNA damage and mitochondrial caspase-dependent apoptosis in cardiomyocytes

To characterize the Dox-induced cell death profile, we performed time response curves (12 to 48 hr) of Dox exposure (1 µM) in neonatal rat ventricular myocytes (NRVM) by flow cytometry analysis. Representative monoparametric histograms of FDA fluorescence and TMRM fluorescence, and representative biparametric cytograms for cell size selection, in the presence or absence of EPAC1 inhibitor, are presented in *Figure 1—figure supplement 1a, b, and c*, respectively. *Figure 1a* shows a time-dependent increase of FDA negative cells, indicative of cell death in association with cell size reduction (*Figure 1b*), loss of ΔΨm indicative of mitochondrial membrane permeabilization (*Figure 1c*) and decreased DNA content (*Figure 1e*). By contrast, Dox did not increase propidium iodide (PI) positive cells percentage, indicator of undamaged cardiomyocytes' plasma membranes, thus excluding necrosis (*Figure 1d*). In addition, the level of the active (cleaved) forms of the mitochondrial-dependent initiator caspase 9 and the executive caspase 3 were increased after Dox treatment (*Figure 1f and*

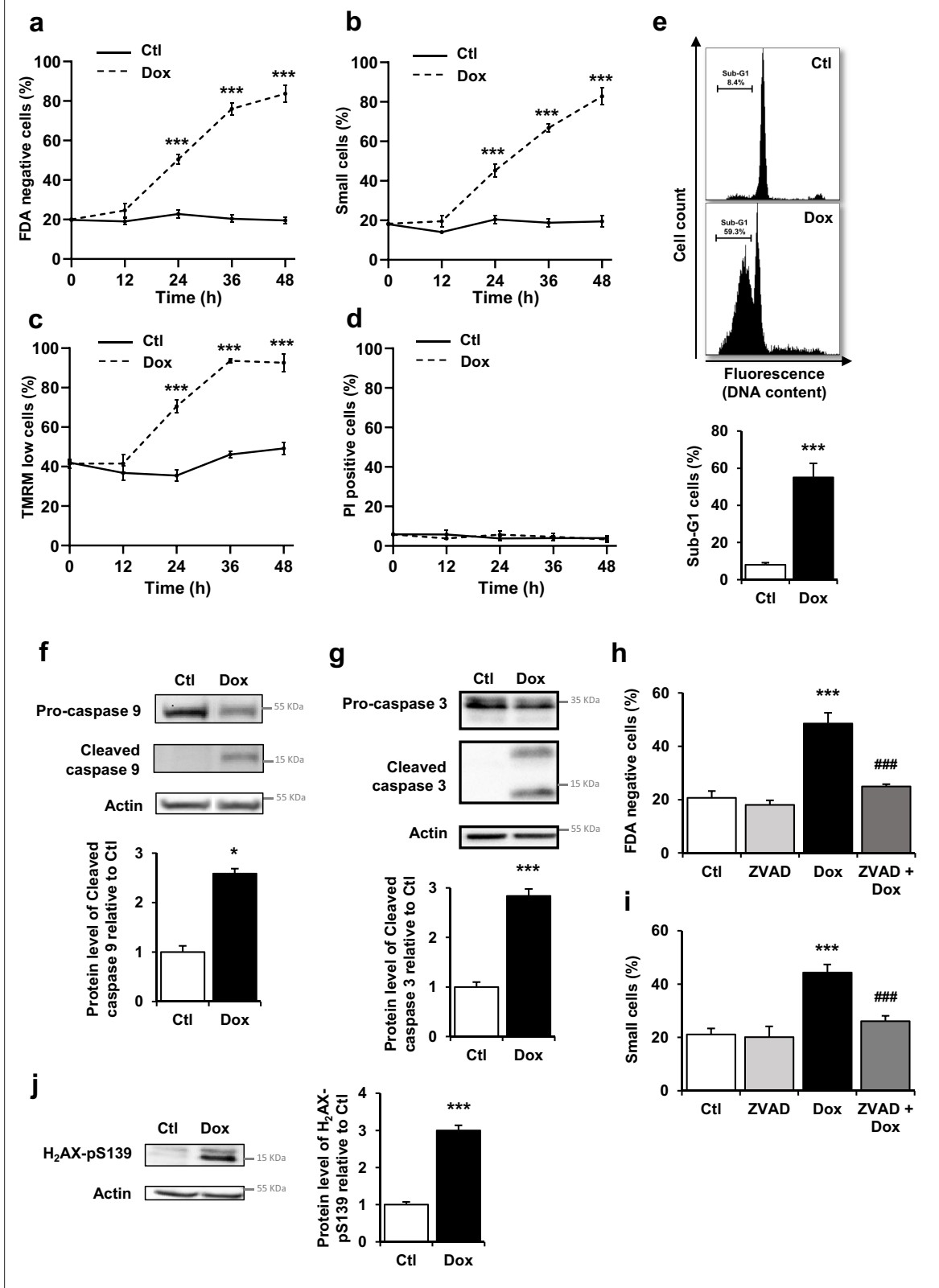

**Figure 1.** Doxorubicin (Dox) induces DNA damages and activates mitochondrial pathway of apoptosis in cardiac myocytes. (**a–d**) Cell death markers were recorded by flow cytometry in neonatal rat ventricular myocyte (NRVM) treated or not with Dox (1 µM) for 12 hr, 24 hr, 36 hr, and 48 hr. Results are presented as mean ± SEM. ***p<0.001 vs. control. (**a**) Percentages of dead cells (FDA negative cells) measured by FDA assay (n=4–12). (**b**) Percentages of small cells obtained by gating the cell population with decrease forward scatter signal (n=4–12). (**c**) Percentages of cells with decreased ΔΨm (TMRM

*Figure 1 continued on next page*

*Figure 1 continued*

low cells) recorded after TMRM staining (n=3–12). (**d**) Necrosis was assessed using propidium iodide (PI) and the percentage of cells with permeabilized plasma membrane (PI positive cells) is presented (n=4–7). (**e**) NRVM were untreated (Ctl) or treated with Dox (1 µM) for 24 hr and the percentage of cells with fragmented DNA (sub-G1 DNA content) was recorded by flow cytometry and is presented as mean ± SEM (n=4). ***p<0.001 vs. control. (**f**, **g**) NRVM were treated or not with Dox (1 µM) for 6 hr and the levels of cleaved-caspase 9 (f) or cleaved-caspase 3 (g) were detected by western blot. Actin was used as a loading control and protein levels relative to Ctl are presented in bar graphs (mean ± SEM, n=3 for caspase 9 and n=11 for caspase 3). *p<0.05, ***p<0.001 vs. control. (**h**, **i**) NRVM were treated or not with Dox (1 µM) ± the general caspase inhibitor ZVAD-fmk (50 µM) for 24 hr. The percentage of dead cells (**h**) and small cells (**i**) is presented (mean ± SEM, n=3–8). ***p<0.001 vs. control, ###p<0.001 vs. Dox alone. (**j**) NRVM were untreated (Ctl) or treated with Dox (1 µM) for 16 hr and the level of the DNA damage marker phosphorylated histone $H_2AX$ ($H_2AX$-pS139) was analyzed by western blot. Actin was used as a loading control. Protein levels relative to Ctl are presented in bar graphs (mean ± SEM, n=4). ***p<0.001 vs. control.

The online version of this article includes the following source data and figure supplement(s) for figure 1:

**Source data 1.** Raw data for *Figure 1*.

**Source data 2.** Uncropped WB for *Figure 1*.

**Source data 3.** Raw WB imaging for *Figure 1*.

**Figure supplement 1.** Doxorubicin (Dox) induced apoptotic cell death in cardiac myocytes.

---

*g*). In the same line, the general caspase inhibitor ZVAD-fmk prevented Dox-induced increase in FDA negative cells and small cells (*Figure 1h and i*), suggesting a caspase-dependent mechanism. DNA damage induced by Dox was assessed by measuring the level of phospho-$H_2AX$ ($H_2AX$-pS$_{139}$, a sensitive marker of DNA double strand breaks) in cultured NRVM at 16 hr to explore the underlying mechanisms prior to the Dox-induced cell death. *Figure 1j* shows that Dox treatment increased the level of $H_2AX$-pS$_{139}$ indicating DNA damage. Altogether, these results demonstrate that Dox induced DNA damage and apoptosis but not necrosis via mitochondrial- and caspase-dependent pathways in NRVM without necrosis induction.

## Dox-triggered cAMP-EPAC1 pathway in cardiac cells

To determine whether EPAC1 is involved in Dox-induced cardiotoxicity, we first analyzed EPAC1 protein level and activity after Dox treatment in NRVM. We observed an up-regulation of EPAC1 protein (*Figure 2a*) by Dox during the acute phase response (3 hr and 6 hr) which decreased when apoptosis is triggered (24 hr). The activity of EPAC1, recorded over time using a pull-down assay of Rap1, showed an increase upon Dox treatment to similar levels as those observed with 10 µM of the EPAC activator 8-CPT (*Figure 2b*). This increase in EPAC1 activity upon Dox treatment was confirmed using a CAMYEL BRET sensor assay (*Figure 2c*), along with an increase of intracellular cAMP concentration, which directly activates EPAC1 (*Figure 2d*). These results show that Dox induced an increase in both EPAC1 expression and activity in NRVM.

The role of EPAC in the Dox-induced DNA damages was assessed by using the EPAC1 pharmacological inhibitor, CE3F4 (*Courilleau et al., 2012*). As shown in *Figure 3a*, CE3F4 (10 µM) decreased $H_2AX$ phosphorylation promoted by Dox, suggesting less DNA double strand breaks. Similarly, EPAC1-specific knockdown by shRNA (shEPAC1, *Figure 3b*, top panel) decreased Dox-induced phosphorylation of $H_2AX$ (*Figure 3b*, lower panel). Moreover, 8-CPT (10 µM) increased Dox-induced $H_2AX$ phosphorylation, while the nonselective EPAC1/EPAC2 inhibitor, ESI-09 (*Zhu et al., 2015*), but not the selective EPAC2 inhibitor, ESI-05 (*Tsalkova et al., 2012*), prevented Dox-induced phosphorylation of $H_2AX$ (*Figure 3c*). These results suggest that the specific inhibition of EPAC1 isoform prevented the formation of Dox-induced DNA double strand breaks.

TopIIβ has a central role in anthracyclines cardiotoxicity since Dox by stabilizing the cleavable complex DNA/TopIIβ generates DNA double strand break (*Moro et al., 2004*). We therefore examined the EPAC and TopIIβ relation by performing a band depletion assay to quantify free TopIIβ protein, not involved in the DNA/TopIIβ complex (*Figure 3d*). Dox decreased the free TopIIβ quantity suggesting cleavable complexes formation, an effect in part prevented by ESI-09, but not ESI-05. Altogether, these results indicate that EPAC1, but not EPAC2, is involved in Dox-induced DNA damage and that pharmacological inhibition of EPAC1 may be useful to prevent Dox-induced DNA/TopIIβ cleavable complex formation and subsequent DNA strand breaks.

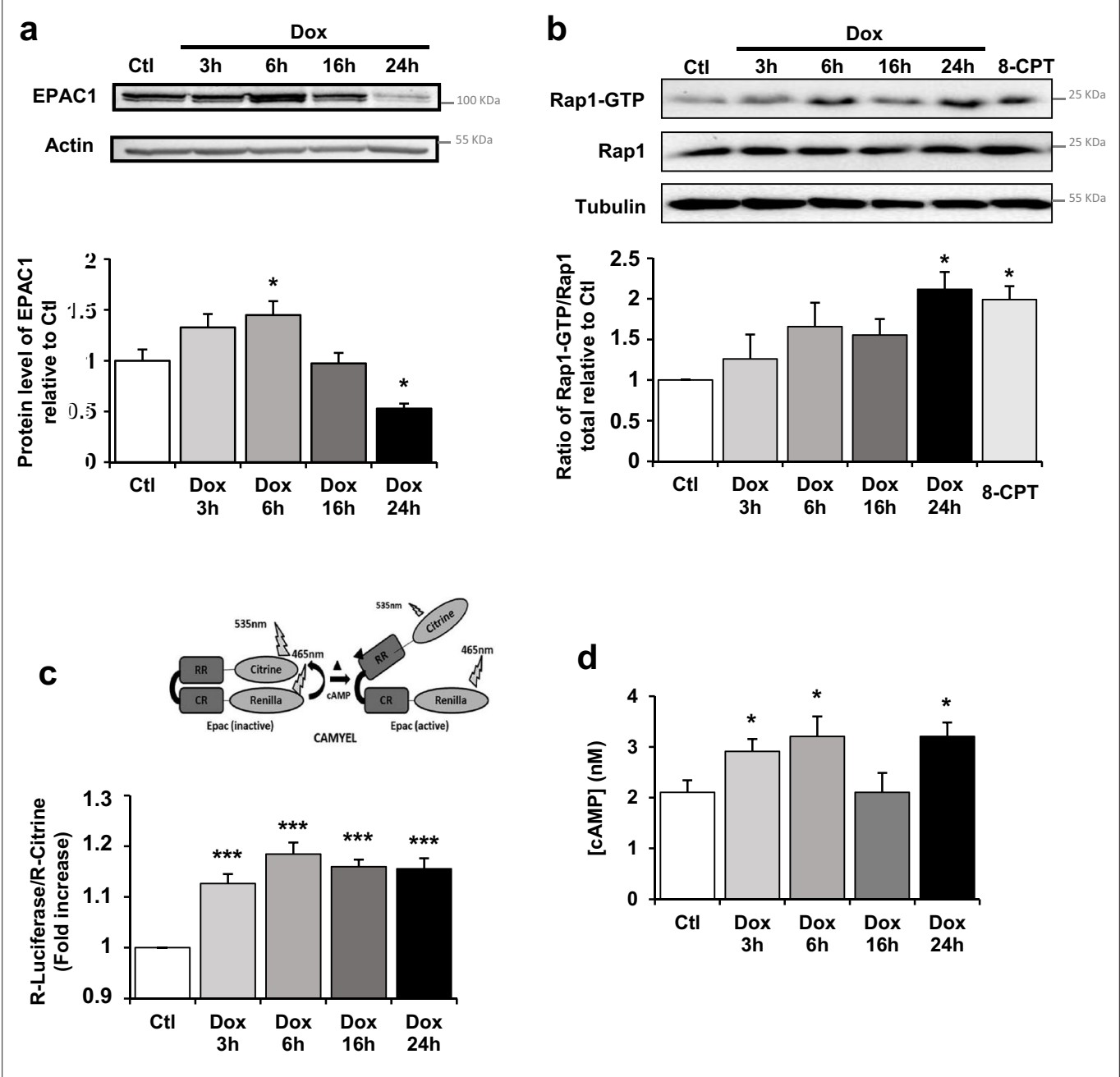

**Figure 2.** Doxorubicin (Dox) modulates cAMP-EPAC1 pathway in cardiac cells. (**a**) Neonatal rat ventricular myocyte (NRVM) were untreated or treated with 1 μM Dox for 3 hr, 6 hr, 16 hr, and 24 hr and the level of EPAC1 protein was detected by western blot. Actin was used as a loading control. Protein levels relative to Ctl are presented in bar graphs (mean ± SEM, n=4–6). *p<0.05 vs. control. (**b**) NRVM were untreated or treated with Dox (1 μM) for 3 hr, 6 hr, 16 hr, and 24 hr or with the EPAC activator 8-CPT (10 μM) for 3 hr. GTP-activated form of RAP1 was analyzed by pull-down assay. RAP1-GTP/RAP1 total ratios relative to Ctl are expressed in bar graphs as mean ± SEM (n=4). *p<0.05 vs. control. (**c–d**) NRVM were untreated or treated with Dox (1 μM) for 3 hr, 6 hr, 16 hr, and 24 hr. (**c**) CAMYEL-based EPAC1 BRET sensor was used to measure EPAC1 activation. The BRET ratio was calculated as the ratio of the Renilla luciferase emission signal to that of citrine-cp (means ± SEM, n=8). ***p<0.001 vs. control. (**d**) The concentration of cAMP (nM) was monitored by cAMP dynamic 2 kit (means ± SEM, n=4). *p<0.05 vs. control.

The online version of this article includes the following source data for figure 2:

**Source data 1.** Raw data for *Figure 2*.

**Source data 2.** Uncropped WB for *Figure 2*.

**Source data 3.** Raw WB imaging for *Figure 2*.

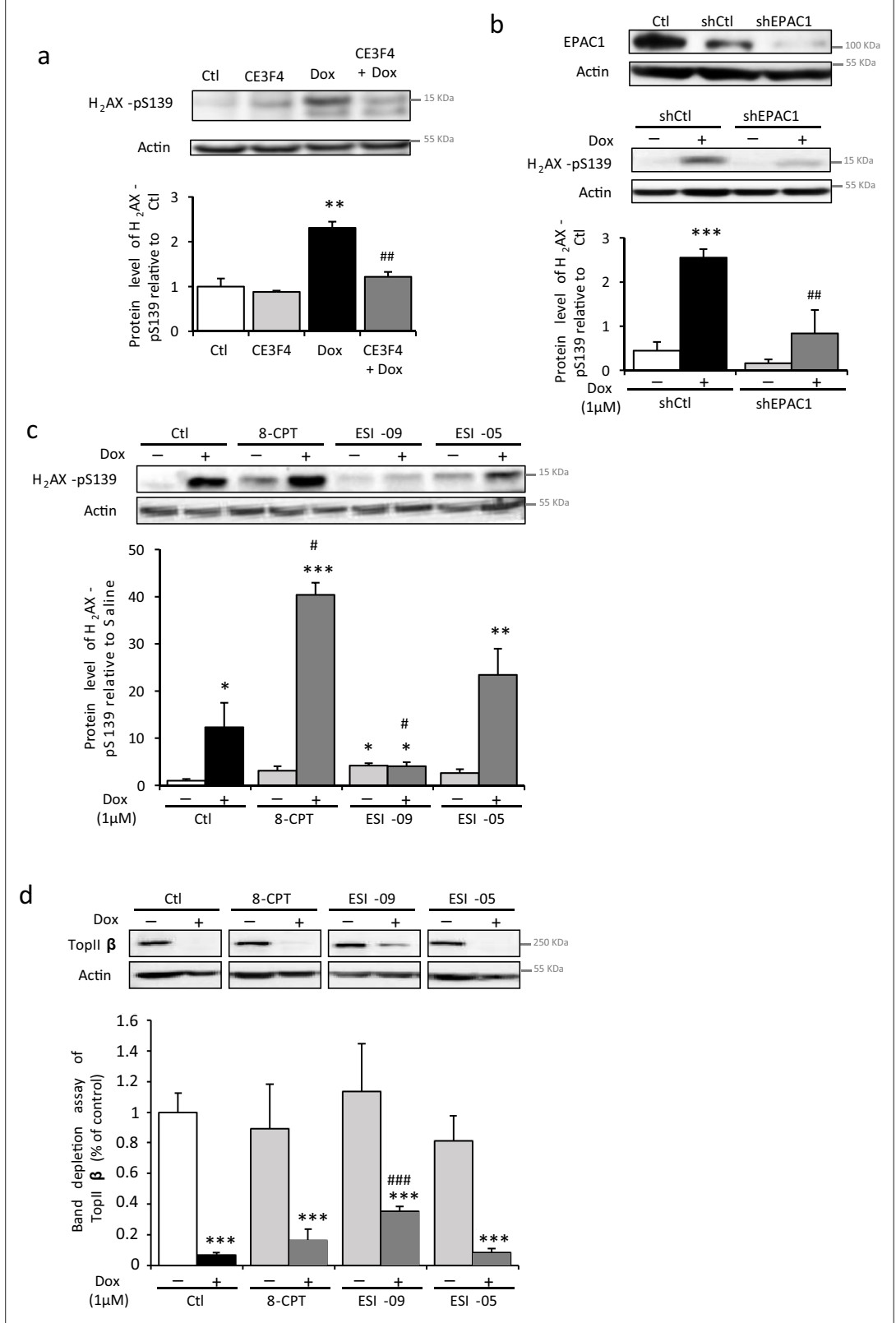

**Figure 3.** Pharmacological inhibition of EPAC1, but not EPAC2, protects cardiomyocytes from doxorubicin (Dox)-induced DNA damage. (**a**) Neonatal rat ventricular myocyte (NRVM) were untreated or treated with Dox (1 µM) ± the specific EPAC1 inhibitor CE3F4 (10 µM) for 16 hr and the level of the DNA damage marker $H_2AX$-pS139 was analyzed by western blot. Actin was used as a loading control. Protein levels relative to Ctl are presented in

*Figure 3 continued on next page*

*Figure 3 continued*

bar graphs (mean ± SEM, n=3–5). **p<0.01 vs. control, ##p<0.01 vs. Dox alone. (**b**) NRVM were transfected with control (shCtl) or EPAC1 (shEPAC1) shRNA for 12 hr before treatment with Dox for 16 hr. The relative levels of EPAC1 and H₂AX-pS139 were measured by immunoblotting with actin as a loading control (mean ± SEM, n=3–5). ***p<0.001 vs. shCtl, ##p<0.01 vs. shCtl + Dox. (**c, d**) NRVM were untreated or treated with Dox (1 µM) and either the EPAC activator 8-CPT (10 µM) or the EPAC1 inhibitor ESI-09 (1 µM) or the EPAC2 inhibitor ESI-05 (1 µM) for 24 hr. (**c**) The level of the DNA damage marker H₂AX-pS139 was analyzed by western blot. Actin was used as a loading control. Protein levels relative to Ctl are presented in bar graphs (mean ± SEM, n=3). *p<0.05, **p<0.01, ***p<0.001 vs. control, #p<0.05 vs. Dox alone. (**d**) The level of free topoisomerase IIβ (TopIIβ) was quantified by band depletion assay. Actin was used as a loading control. Protein levels relative to Ctl are presented in bar graphs (mean ± SEM, n=3–5). ***p<0.001 vs. control, ###p<0.001 vs. Dox alone.

The online version of this article includes the following source data for figure 3:

**Source data 1.** Raw data for *Figure 3*.

**Source data 2.** Uncropped WB for *Figure 3*.

**Source data 3.** Raw WB imaging for *Figure 3*.

## EPAC1 inhibition reduced Dox-induced mitochondrial apoptotic pathway

As metabolism and mitochondrial disorders are an important pattern of Dox cardiotoxicity downstream of TopIIβ signaling alterations, we next investigated whether EPAC1 inhibition protects NRVM from Dox-induced mitochondrial alterations. As shown in *Figure 4a*, CE3F4 reduced the Dox-evoked dissipation of ΔΨm. Prolonged opening of the MPTP is one of the mechanisms known to initiate mitochondrial membrane permeabilization leading to ΔΨm loss. MPTP opening was thus analyzed by flow cytometry using calcein-cobalt assay. Dox treatment elicited opening of the MPTP (calcein negative cells), which was decreased by EPAC1 inhibition (*Figure 4b*). We next evaluated the mitochondrial function under Dox ± CE3F4 treatment by measuring the activity of the complex I and IV by cytochrome *c* oxidase and ubiquinone oxidoreductase activities measurements. EPAC1 inhibition prevented the Dox-induced complex I and IV activity decrease (*Figure 4c and d*). Altogether, these data show that EPAC1 inhibition by CE3F4 protects NRVM from Dox-induced mitochondrial dysfunction.

We also investigated whether EPAC1 is involved in Dox-induced apoptosis. The Dox-induced increase in the level of active caspase 9 (*Figure 4e*), caspase 3 (*Figure 4f*), the percentages of FDA negative cells (*Figure 4g*) and small cells (*Figure 4h*) were significantly reduced by CE3F4, while ESI-05 did not (*Figure 4j*). Interestingly, the CE3F4 protection was comparable to that observed with the clinically approved dexrazoxane (*Figure 4g*). The micrographs shown in *Figure 4i* depict the morphology of NRVM in the presence or absence of Dox and CE3F4. While many Dox-treated cardiomyocytes aggregated and rounded up, suggestive of dying cells, most CE3F4 supplemented cells exhibited normal shape. The protective activity of CE3F4 was also tested in a more differentiated model of adult rat ventricular myocytes (ARVM). As in NRVM, CE3F4 protected ARVM from Dox-induced cell death (*Figure 4k*). These data demonstrate that pharmacological inhibition of EPAC1, but not EPAC2, attenuates Dox-induced cell death both in NRVM and ARVM, suggesting that there is an EPAC isoform specificity in Dox-induced cardiotoxicity.

## Dox-induced cardiotoxicity was prevented in EPAC1 KO mice

Since EPAC1 inhibition showed a strong protection against Dox treatment on various parameters (cell death, mitochondria, DNA damages) in NRVM and ARVM, the obvious question was whether EPAC1 inhibition could also provide cardioprotection *in vivo*. To test this hypothesis, we used mice with global *Rapgef3* deletion (EPAC1 KO). Dox or saline treatment was injected to both wild type (WT) and EPAC1 KO mice and analyzed 15 weeks after the last injection, when DCM was established (*Llach et al., 2019*). A growth delay was observed in Dox-treated mice, both in WT and EPAC1 KO mice in comparison to saline-treated mice (*Figure 5—figure supplement 1*). However, EPAC1 KO mice exhibited a preserved cardiac function after Dox treatment as evidenced by unaltered ejection fraction (EF) and left ventricle end-diastolic volume (LVEDV) (*Figure 5a, b, c, f, and g*) indicating a complete prevention of DCM development. Furthermore, WT heart samples showed that the

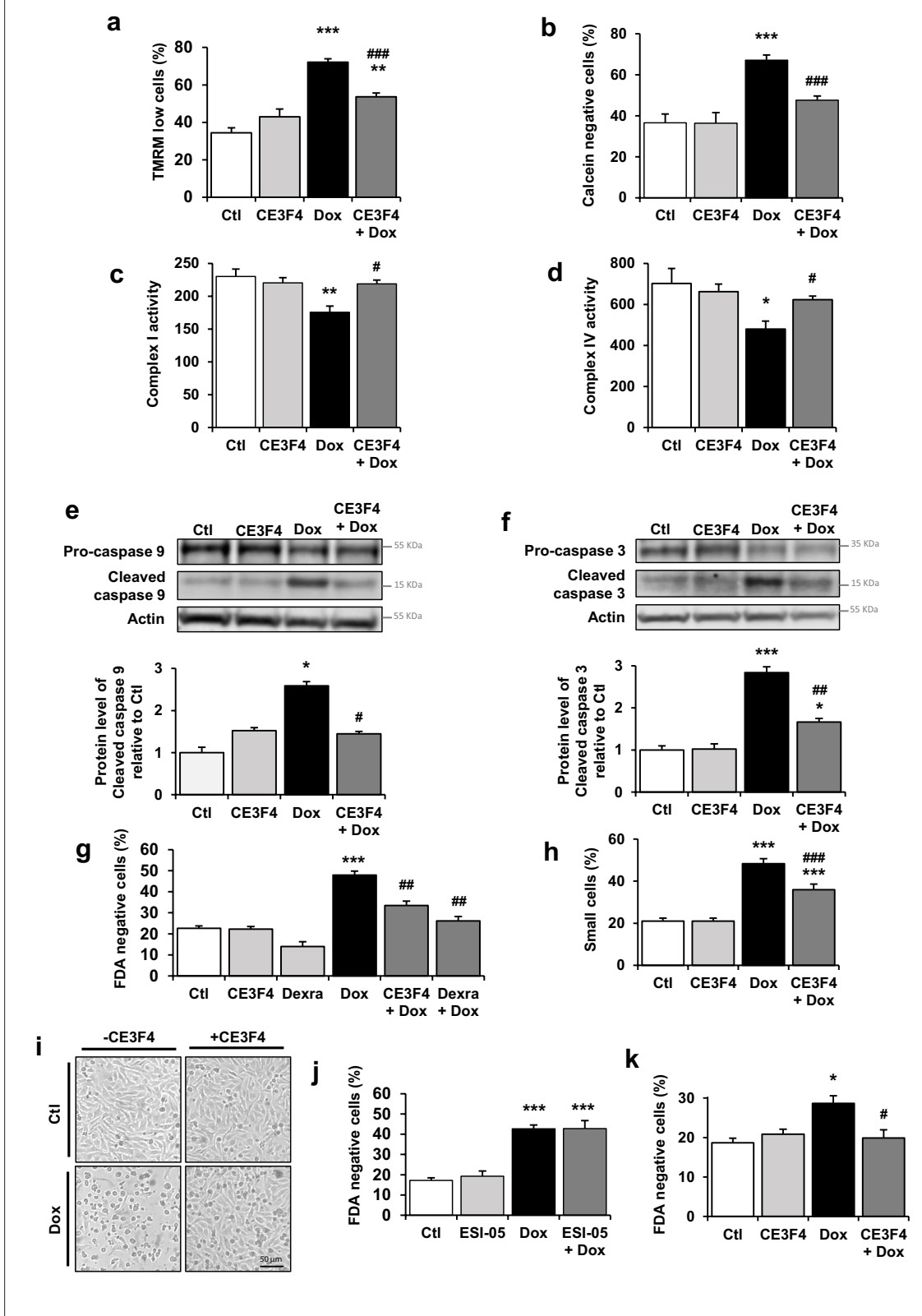

**Figure 4.** EPAC1 inhibition reduces doxorubicin (Dox)-induced mitochondrial apoptotic pathway. (**a, b**) Neonatal rat ventricular myocyte (NRVM) were untreated or treated with Dox (1 μM) ± CE3F4 (10 μM) for 24 hr and analyzed by flow cytometry. Results are expressed as means ± SEM. **p<0.01, ***p<0.001 vs. control, ###p<0.001 vs. Dox alone. (**a**) TMRM staining was used to assess the percentage of cells with decreased ΔΨm (as TMRM low cells) (n=10). (**b**) Calcein-cobalt assay was used to record MPTP opening (as calcein negative cells) (n=3–4). (**c, d**) NRVM were untreated or treated with

*Figure 4 continued on next page*

*Figure 4 continued*

Dox (1 µM) ± CE3F4 (10 µM) for 16 hr. Activity of the mitochondrial respiratory complex I (n=4) (**c**) and complex IV (n=3) (**d**) is presented in the bar graphs as means ± SEM. *p<0.05, **p<0.01 vs. control, #p<0.05 vs. Dox alone. (**e**, **f**) NRVM were untreated or treated with Dox (1 µM) ± CE3F4 (10 µM) for 6 hr and the levels of cleaved-caspase 9 (**e**) and cleaved-caspase 3 (**f**) were detected by western blot. Actin was used as a loading control and protein levels relative to Ctl are presented in bar graphs (means ± SEM, n=3 for caspase 9 and n=7–11 for caspase 3). *p<0.05, ***p<0.001 vs. control, #p<0.05, ##p<0.01 vs. Dox alone. (**g**, **h**) NRVM were treated or not with Dox (1 µM) ± CE3F4 (10 µM) ± dexrazoxane (200 µM) for 24 hr and analyzed by flow cytometry. Results in bar graphs are expressed as mean ± SEM. ***p<0.001 vs. control, ##p<0.01, ###p<0.001 vs. Dox alone. (**g**) Percentage of dead cells (FDA negative cells) (n=3–7). (**h**) Percentage of small cells (n=8). (**i**) Representative micrographs of NRVM untreated or treated with Dox (1 µM) ± CE3F4 (10 µM) for 24 hr. (**j**) The percentage of dead cells (FDA negative cells) was assessed in NRVM left untreated or treated with Dox (1 µM) ± the specific EPAC2 inhibitor ESI-05 (1 µM) and presented as means ± SEM (n=5). ***p<0.001 vs. control, ###p<0.001 vs. Dox alone. (**k**) Freshly isolated adult rat ventricular myocytes (ARVM) were treated or not with Dox (1 µM) ± CE3F4 (10 µM) for 24 hr and cell death was determined. The percentage of dead cells (FDA negative cells) is presented in bar graphs (means ± SEM, n=4–5). *p<0.05 vs. control, #p<0.05 vs. Dox alone.

The online version of this article includes the following source data for figure 4:

**Source data 1.** Raw data for *Figure 4*.

**Source data 2.** Uncropped WB for *Figure 4*.

**Source data 3.** Raw WB imaging for *Figure 4*.

expression of EPAC1 was increased 6 weeks after Dox treatment and strongly decreased at 15 weeks, when DCM is detected (*Figure 5d*).

To verify that the cardioprotection conferred by *Rapgef3* deletion took place at the level of the cardiomyocyte, ventricular myocytes were isolated from WT and EPAC1 KO mice 15 weeks after last i.v. injection and loaded with fluorescence $Ca^{2+}$ dye Fluo-3 AM to record $[Ca^{2+}]_i$ transients and cell shortening in electrically stimulated cardiomyocytes. *Figure 6* shows that while cardiomyocytes isolated from Dox-treated WT mice had a reduced unloaded cell contraction (cell shortening, *Figure 6a*), a reduced $[Ca^{2+}]_i$ transient amplitude (peak $F/F_0$, *Figure 6b*), and a decreased sarcoplasmic reticulum (SR) $Ca^{2+}$ load (*Figure 6c*), all these alterations were absent in myocytes isolated from Dox-treated EPAC1 KO mice. Surprisingly, $[Ca^{2+}]_i$ transient amplitude and SR $Ca^{2+}$ load were significantly increased in Dox-treated EPAC1 KO mice as compared to saline (*Figure 6b and c*). This was reminiscent of the compensatory response seen in WT mice after 6 weeks of Dox treatment, where both parameters were similarly increased (*Llach et al., 2019*). This compensatory response to Dox was transient in WT mice (*Llach et al., 2019*) but may persist in EPAC1 KO mice because of the protective effect of *Rapgef3* deletion on the decompensation phase. *Figure 6d* also shows that *Rapgef3* deletion prevented the Dox-induced decrease in SERCA2A protein expression, which in WT mice was accompanied by a slowdown in the relaxation of the $[Ca^{2+}]_i$ transients (*Figure 6e*). These results demonstrate a complete prevention of Dox-induced toxicity at the level of the cardiomyocyte by genetic deletion of *Rapgef3*.

## EPAC1 inhibition enhanced the cytotoxic effect of Dox in human cancer cell lines

While the above experiments tend to suggest that EPAC1 may be a potential therapeutic target to limit Dox-induced cardiotoxicity, it is important to verify that EPAC1 inhibition does not compromise the anticancer efficacy of Dox. For that, we tested the effect of CE3F4 on two cancer cell lines, MCF-7 human breast cancer and HeLa human cervical cancer, which are derived from tumors usually treated with Dox. Dox dose response curves ± CE3F4 (10 µM) were performed and cell death (FDA) was measured (after 24 hr) using flow cytometry. Dox induced cell death in both cancer cell lines in a dose-dependent manner (*Figure 7*). Interestingly, dead cells percentage was significantly increased in both cell lines with CE3F4 (*Figure 7a and b*). Therefore, EPAC1 inhibition not only protects cardiac cells from Dox-induced toxicity but also enhances the anticancer efficacy of this anthracycline.

## Discussion

Dox is a potent chemotherapeutic agent used in clinic to treat a wide range of cancers but this anthracycline is also well known for its cardiotoxic side effects. Since decades, Dox-induced toxicity is considered to occur mostly through DNA damage, ROS generation, energetic distress, and cell death (apoptosis, necrosis, etc.). However, non-canonical/alternative pathways are nowadays emerging and Dox-induced cardiotoxicity was recently proposed to occur through mechanisms other than those

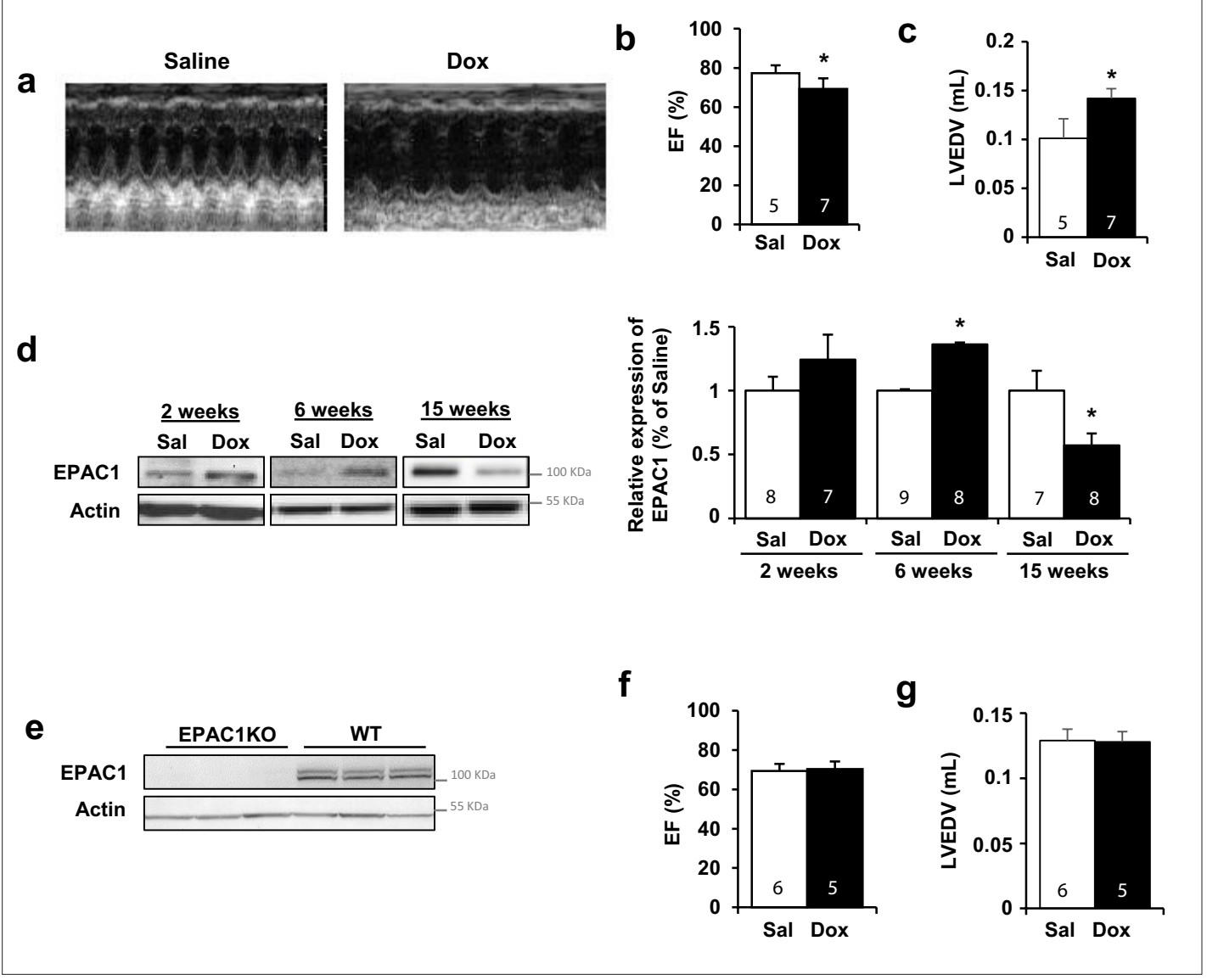

**Figure 5.** Doxorubicin (Dox)-induced cardiotoxicity was prevented in EPAC1 knock-out (KO) mice. WT (**a–d**) or EPAC1 KO mice (**e–g**) were injected (i.v.) three times with saline solution (Sal) or Dox at 4 mg/kg for each injection (12 mg/kg cumulative dose). Echocardiographic analysis was performed at 15 weeks after the last injection. (**a**) Representative echocardiography images. (**b**) Ejection fraction (EF) and (**c**) left ventricle end-diastolic volume (LVEDV) are presented as means ± SEM. *$p < 0.05$ vs. saline. (**d**) The protein level of EPAC1 was determined by western blot at 2, 6, and 15 weeks after the last injection. Actin was used as a loading control. Relative protein levels are presented in bar graphs. *$p < 0.05$ vs. saline. (**e**) Absence of EPAC1 protein was verified in EPAC1 KO mice by western blot. Actin was used as a loading control. (**f**) EF and (**g**) LVEDV in EPAC1 KO mice are presented as means ± SEM. In (**b**, **c**, **d**, **f**, and **g**), the number of mice is indicated in the corresponding bargraph.

The online version of this article includes the following source data and figure supplement(s) for figure 5:

**Source data 1.** Raw data for *Figure 5*.

**Source data 2.** Uncropped WB for *Figure 5*.

**Source data 3.** Raw WB imaging for *Figure 5*.

**Figure supplement 1.** Body weight of mice treated with doxorubicin (Dox).

**Figure 5-figure supplement 1-source data 1.** Raw data for *Figure 5—figure supplement 1*.

mediating its anticancer activity (*Zhang et al., 2012*; *Li et al., 2018*; *Zhang et al., 2019*). Moreover, no widely accepted therapeutic strategies to minimize Dox-induced cardiac injury have been established. The iron chelator dexrazoxane is the only FDA- and EMEA-approved cardioprotector despite limited and still controversial application (*Tebbi et al., 2007*). Here, we demonstrated that

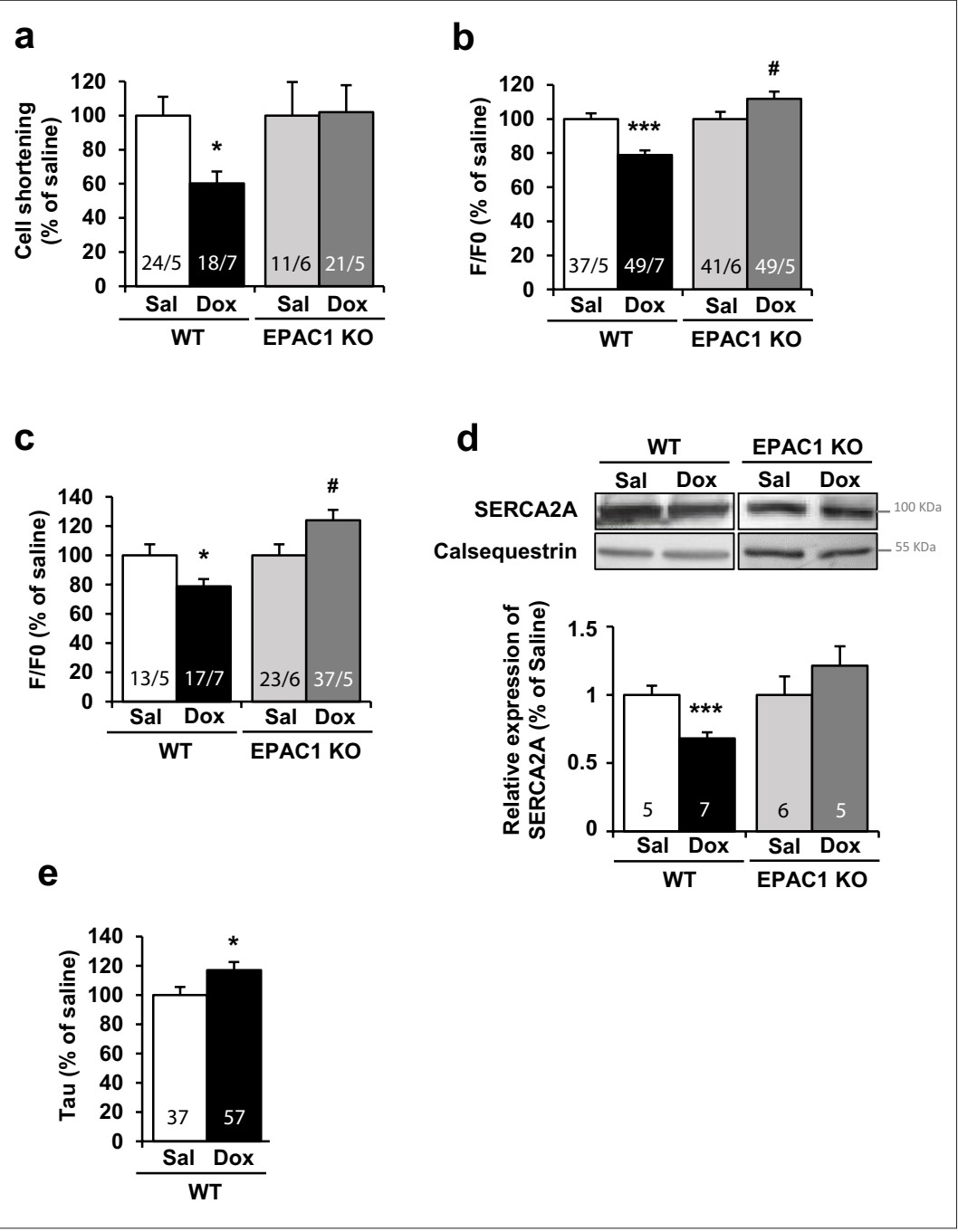

**Figure 6.** Doxorubicin (Dox)-induced cardiotoxicity was prevented in EPAC1 knock-out (KO) mice. WT and EPAC1 KO mice were injected (i.v.) three times with saline solution (Sal) or Dox at 4 mg/kg for each injection (12 mg/kg cumulative dose). Ventricular cells were isolated from control and treated mice 15 weeks after last i.v. injection and loaded with fluorescence $Ca^{2+}$ dye Fluo-3 AM allowing to measure (**a**) cell shortening, (**b**) calcium transient amplitude, and (**c**) sarcoplasmic reticulum calcium release (upon application of 10 mM caffeine) by confocal microscopy. Normalized values are presented as mean ± SEM. *$p < 0.05$, ***$p < 0.001$ vs. WT saline, #$p < 0.05$, ##$p < 0.01$ vs. EPAC1 KO saline. The number of animals and cells is indicated in the bars of the graphs. (**d**) The level of SERCA2A protein was measured by western blot. Calsequestrin was used as a loading control. Relative protein levels are presented as mean ± SEM. ***$p < 0.001$ vs. WT saline. The number of saline or Dox-treated mice is indicated in the bars of the graphs. (**e**) Histograms of normalized mean ± SEM values of relaxation time constant (Tau) of calcium transients in Sal and Dox. The number of animals and cells is indicated in the bars.

The online version of this article includes the following source data for figure 6:

*Figure 6 continued on next page*

*Figure 6 continued*

**Source data 1.** Raw data for *Figure 6*.

**Source data 2.** Uncropped WB for *Figure 6*.

**Source data 3.** Raw WB imaging for *Figure 6*.

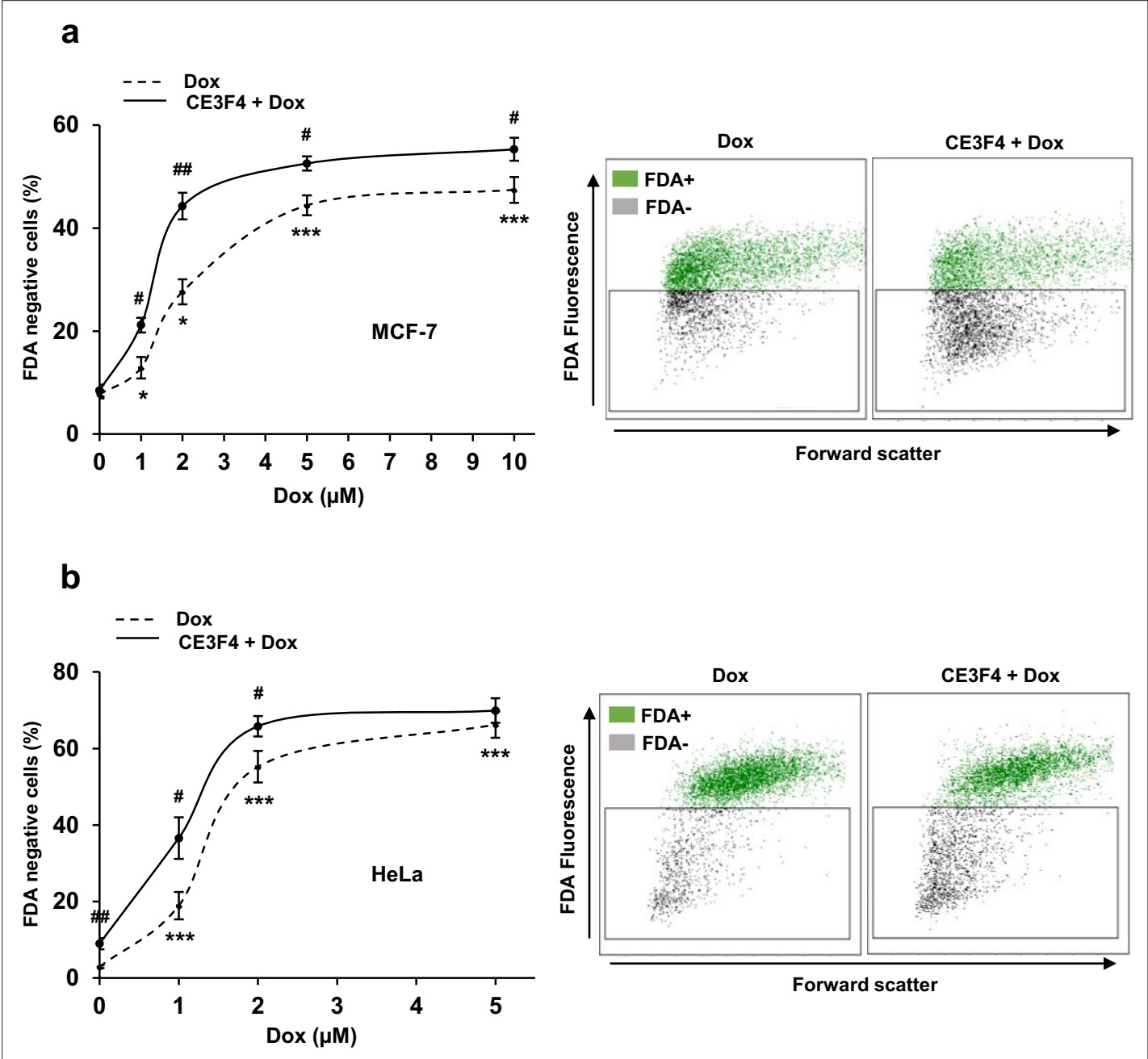

**Figure 7.** EPAC1 inhibition enhances doxorubicin (Dox)-induced cytotoxicity in various human cancer cell lines. Human MCF-7 breast cancer cells (**a**) and HeLa cervical cancer cells (**b**) were untreated or treated with the indicated doses of Dox ± CE3F4 (10 µM) for 24 hr. Cell death was measured by flow cytometry after FDA staining. Results presented in graphs are expressed as the percentage of dead cells (FDA negative cells) (means ± SEM, n=5 for MCF-7 and n=8–9 for HeLa, *p<0.05, ***p<0.001 vs. control, #p<0.05, ##p<0.01 vs. Dox alone). Representative biparametric cytograms showing FDA fluorescence vs. forward scatter (cell size) are presented.

The online version of this article includes the following source data for figure 7:

**Source data 1.** Raw data for *Figure 7*.

pharmacological inhibition (CE3F4) and the use of EPAC1 KO mice prevented Dox-induced cardio-toxicity *in vitro* and *in vivo*. We report an up-regulation of expression and activity of EPAC1 following Dox treatment. Moreover, specific EPAC1 inhibition with CE3F4, but not EPAC2 inhibition with ESI-05, reduced DNA damage, mitochondrial alterations, and apoptosis elicited by Dox in cardiomyocytes, indicating that EPAC1, but not EPAC2, inhibition protects the heart against Dox-induced cardiotox-icity. This was confirmed *in vivo* in EPAC1 KO mice. Moreover, EPAC1 inhibition by CE3F4 not only protected cardiac cells, but also increased the toxicity of Dox against human cancer cell lines (MCF-7 human breast cancer and HeLa human cervical cancer). Therefore, these results strongly suggest that inhibition of EPAC1 could be a promising strategy to prevent heart damage in anthracyclines-treated cancer patients.

At the molecular level, there is evidence that Dox and EPAC signaling pathways share common features and downstream effectors (*Sag et al., 2011*; *Riganti et al., 2008*; *Huelsenbeck et al., 2012*). However, no direct link between these two pathways has been reported so far. Our *in vitro* kinetic analysis revealed that Dox induces an increase of EPAC1 expression in the first hours after treatment (3–6 hr), followed by a down-regulation (16–24 hr). Despite this drop, EPAC activity remained elevated during the 24 hr of Dox treatment. This was confirmed by the progressive increase of the active form Rap-GTP, the direct EPAC effector, and was most likely due to the observed elevation of cAMP concentration in cardiomyocytes during the first 24 hr following Dox treatment. A subsequent decrease of EPAC1 protein level was observed which likely represents a protective cellular response, since we found that chemical or genetic EPAC1 inhibition was cardioprotective.

We show that Dox induces a mitochondrial caspase-dependent apoptosis characterized by DNA damage, $Ca^{2+}$-dependent MPTP opening and mitochondrial membrane permeabilization, caspase activation, nuclear fragmentation, and cell size reduction which is consistent with previous report (*Zhang et al., 2009*). However, by contrast to other studies (*Lim et al., 2004*; *Lebrecht and Walker, 2007*), 1 µM Dox did not induce necrosis in NRVM (measured by PI staining), which was only observed when higher doses of Dox (up to 10 µM) were used (data not shown), indicating that induction of cardiac necrosis by Dox may be dose-dependent.

The role of EPAC1 in cardiomyocyte apoptosis is still unclear owing to controversial reports (*Mangmool et al., 2015*; *Okumura et al., 2014*; *Suzuki et al., 2010*; *Fazal et al., 2017*). Indeed, EPAC1 has anti- or pro-apoptotic effect in the heart, depending on the type of cardiomyopathy and its induction mode. For instance, EPAC1 was reported to be cardioprotective and to participate in antioxidant and anti-apoptotic effects of exendin-4, an agonist of the glucagon-like peptide-1 receptor (*Mangmool et al., 2015*). By contrast, EPAC1 KO mice showed no cardiac damage induced by pressure overload, chronic catecholamine injection, or ischemia-reperfusion (*Okumura et al., 2014*; *Fazal et al., 2017*), suggesting a deleterious effect of EPAC1. This deleterious role of EPAC1 is also found in other tissues as *Rapgef3* deletion or its inhibition by ESI-09 reduces nerve injury and inflammation, leading to allodynia upon anticancer drug paclitaxel treatment (*Singhmar et al., 2018*). The protective/detrimental action of EPAC1 is thus not completely understood and might depend on the tissues and the pathophysiological mechanisms of the disease. Here, using pharmacological (CE3F4) and genetic approaches, we found that EPAC1 inhibition is cardioprotective in the context of Dox-induced apoptosis and cardiotoxicity.

Mechanistically, we demonstrated that EPAC1 inhibition prevents MPTP opening induced by Dox. Consistent with our observation, MPTP opening is known to result in mitochondrial membrane permeabilization, pro-apoptotic factors release, activation of caspases, and cell dismantling (*Childs et al., 2002*; *Deniaud et al., 2008*). Recently, it has been demonstrated that EPAC1 KO mice are protected against myocardial ischemia/reperfusion injury (*Fazal et al., 2017*). The EPAC1 activation leads to increased mitochondrial $Ca^{2+}$ overload, which in turn promotes MPTP opening, pro-apoptotic factor release, caspase activation, and cardiomyocyte death. The demonstration that EPAC1 inhibition prevents Dox-induced MPTP opening and cardiomyocyte death suggests that EPAC1 may play a similar deleterious role in Dox-associated cardiotoxicity and ischemia/reperfusion injury and that MPTP is a major downstream effector in the cardiac cell death signaling cascade regulated by EPAC1.

We found that Dox elicited mitochondrial dysfunction in cardiomyocytes, in agreement with previous report (*Carvalho et al., 2014*). Indeed, $\Delta\Psi m$ was up-regulated together with a strong decrease of the activity of the mitochondrial respiratory complex I and IV in cardiomyocytes exposed to Dox. Importantly, inhibition of EPAC1 by CE3F4 counteracted the mitochondrial alterations induced

by Dox. These results could be correlated with a previous study showing that during vascular injury, mitochondrial fission and cell proliferation were suppressed by inhibition of EPAC1 in vascular smooth muscle cells (*Wang et al., 2016*). Therefore, inhibiting EPAC1 can help prevent Dox cardiotoxicity as observed in vascular proliferative diseases.

One of the major mechanisms of Dox-induced cardiomyocyte death and mitochondria reprogramming involves TopIIβ. This gyrase is an enzyme that regulates DNA winding by the formation of DNA/TopII cleavable complexes and generation of DNA double strand breaks (*Corbett and Berger, 2004*). Thus, by its important role in replication and transcription, TopIIβ is of particular interest in cancer therapy (*Nitiss, 2009*). Recently, on the cardiac side, a new paradigm in Dox-induced cardiotoxicity has been put forward, conferring a central role to TopIIβ. Indeed, TopIIβ is required for ROS generation, DNA damage, and cell death induction, the three canonical mechanisms involved in Dox side effects (*Zhang et al., 2012*). Our results showed that EPAC1 inhibition prevents Dox-induced formation of DNA/TopIIβ cleavable complexes, as evidenced by higher protein level of free TopIIβ, suggesting that EPAC1 inhibition blocks deleterious Dox effects in part through the TopIIβ pathway.

Recent data have reported that EPAC1 could be essential to modulate cancer development and metastasis formation (*Kumar et al., 2018*). In addition, inhibition of EPAC1 was proposed as a therapeutic strategy for the treatment of cancers such as melanoma, pancreatic, or ovarian cancers (*Almahariq et al., 2016*; *Baljinnyam et al., 2014*; *Baljinnyam et al., 2011*; *Gao et al., 2016*). For instance, the genetic deletion of EPAC1 or the *in vivo* EPAC1 inhibition by ESI-09 decreases migration and metastasis of pancreatic cancer cells (*Almahariq et al., 2015*). Of note, in breast cancer cells, which are generally sensitive to Dox therapy, EPAC1 inhibition was shown to inhibit cell migration and induce apoptosis (*Kumar et al., 2017*). Here, we found that inhibition of EPAC1 by CE3F4 increased Dox-induced cell death in two human cancer cell lines, including breast cancer cells. Therefore, our data show that EPAC1 inhibition not only protects cardiac cells from Dox-induced toxicity but also enhances the sensitivity of cancer cells to Dox. Similarly, PI3Kγ blockade (*Li et al., 2018*) or the novel agent named biotin-conjugated ADTM analog (BAA) (*Zhang et al., 2019*) were shown to prevent Dox-induced cardiotoxicity and to synergize with its antitumor activity against breast cancer. Therefore, the identification of new promising chemical compound, such as BAA or CE3F4, could be the starting point to the next therapeutic treatments that would protect patients from acute and chronic cardiotoxicity of Dox without altering its cytotoxicity toward cancer cells.

In conclusion, we propose EPAC1 as a new regulator of Dox-induced toxicity in cardiac cells. In response to Dox, the pharmacological inhibition of EPAC1 by CE3F4 recapitulated EPAC1 KO phenotype, suggesting the potential therapeutic efficacy of this EPAC1 inhibitor to alleviate Dox-associated side effects in the heart, while maintaining or enhancing anticancer effect. Thus, EPAC1 inhibition represents a promising therapeutic strategy both to prevent Dox cardiotoxicity and to enhance its antitumoral activity.

## Methods

### Animal study

All animal experiments were conducted in line with the French/European Union Council Directives for the laboratory animals care 86/609/EEC (MESRI 18927 authorization).

### Model

Twelve-week-old C57BL6 (wild type littermate [WT] or *Rapgef3* knock-out [EPAC1 KO]; *Laurent et al., 2015*) male mice received i.v. (tail vein) NaCl (0.09%) or Dox (4 mg/kg, three times at 3 days' intervals, cumulative dose of 12 mg/kg) in order to mimic human therapeutic regimen and the induction of a moderated DCM without inducing death (*Desai et al., 2013*). Since sex differences have been reported in Dox-induced cardiotoxicity (*Moulin et al., 2015*), only male animals were used here. Cardiac function was measured at 2, 6, and 15 weeks post-injections. Ventricular myocytes were isolated at the same time points for Ca$^{2+}$ handling and western blot analysis as previously detailed (*Llach et al., 2019*).

## Echocardiography

Mouse heart physiological parameters were measured by trans-thoracic echocardiography (Vivid 9, General Electric Healthcare) at 15 MHz, under isoflurane anesthesia (3%). LVEDV and EF were imaged (two-dimensional mode followed by M-mode). LVEDV was calculated with the Teichholz formula.

## Cell culture

### Dissociation of NRVM and ARVM

NRVM were isolated from 1- to 3-day-old Sprague-Dawley rats (40 pups per dissociation) (Janvier, Le Genest-Saint-Isle, France). Ventricles were digested with collagenase A (Roche, Meylan, France) and pancreatin (Sigma-Aldrich, St Quentin Fallavier, France), separated by a Percoll gradient and plated (DMEM/medium 199 [4:1]) (Thermo Fisher Scientific, Les Ulis, France). Langendorff method was used to isolate ARVM as previously described (*Rochais et al., 2004*).

### Human cancer cell lines

MCF-7 and HeLa cell lines were provided by C Leuter from the Institut Gustave Roussy (Villejuif, France). Cell lines were cultured in DMEM + 10% fetal bovine serum and antibiotics. Cell lines have been authenticated using STR profiling and mycoplasma testing was performed routinely.

### Treatments

Dox (2 mg/mL) and dexrazoxane were obtained from ACCORD (central pharmacy of Institut Gustave Roussy, Villejuif, France) and Sigma (St Quentin Fallavier, France), respectively. The *in vitro* Dox exposure of isolated cardiomyocytes was of 1–10 µM in concordance with literature recommendation (*Tokarska-Schlattner et al., 2006*) and extended to match tumor cell lines exposure requirements.

8-(4-Chloro-phenylthio)-2'-*O*-methyladenosine-3'5'cyclic monophosphate (8-CPT), ESI-09, and ESI-05 were from Biolog Life Science Institute (Bremen, Germany). ZVAD-fmk was from Bachem (Bubendorf, Switzerland). EPAC1 specific inhibitor (R)-CE3F4 was provided by Dr Ambroise (CEA, Gif-sur-Yvette, France) (*Courilleau et al., 2013*).

### Transfections

Cells were transfected with the GST fusion protein construct or the BRET-based cAMP construct (1 µg) using Lipofectamine 2000 (Invitrogen Life Technologies, Saint-Aubin, France) in Opti-MEM medium.

### Infections

NRVM were incubated for 12 hr with recombinant shRNA encoding shCtl (Welgen[INC], Worcester, MA, USA, Dr K Luo) or shEPAC1 (Hôpital universitaire de Nantes, FRANCE, Dr C Darmon) adenoviruses (MOI 500).

## Cell death profiling

Cell death and apoptotic markers were recorded by flow cytometry (FC500, Beckman Coulter, Villepinte, France). Cell viability was analyzed by FDA (fluorescein diacetate) staining (0.2 µg/mL, 10 min) in which FDA negative cells were dead cells and those with decreased forward scatter signal counted as small cells entering the apoptotic process. Mitochondrial membrane potential ($\Delta \Psi$m) was monitored using the cationic dye TMRM (10 nM, 20 min). TMRM low cells were cells with decreased $\Delta \Psi$m. Necrosis was evaluated by staining cells with 10 µg/mL of PI. PI positive cells were necrotic cells. DNA fragmentation (sub-G1 DNA content) was estimated by measuring DNA content after overnight permeabilization (70% ethanol) and incubation (4°C, 24 hr) with PI (50 µg/mL) and RNAse A (250 µg/mL). Mitochondrial permeability transition pore (MPTP) opening was assessed by calcein-cobalt assay as previously described (*Deniaud et al., 2008*).

## Western blot analysis

Proteins were separated by SDS-PAGE (10–15%) and transferred to a PVDF membrane, which were incubated (overnight, 4°C) with antibodies: anti-EPAC1 (1/500), anti-caspase 3 (1/500), anti-caspase 9 (1/1000), anti-RAP1 (1/1000), and anti-H$_2$AX-pS$_{139}$ (1/1000) from Cell Signaling Technology (Saint-Cyr-L'Ecole, France), anti-TopIIβ (1/1000) from Abcam (Paris, France), anti-SERCA2A (1/1000), and

anti-actin (1/50,000) from Santa Cruz Biotechnology (Heidelberg, Germany), anti-calsequestrin (1/2500) from Thermo Fisher Scientific (Les Ulis, France), and anti-tubulin (1/1000) from Sigma (St Quentin Fallavier, France). Proteins were detected on the iBright FL1000 Imager (Thermo Fisher Scientific, Les Ulis, France) with ECL and protein band intensity was calculated using ImageJ software and normalized by actin.

## Measurement of EPAC1 activity and cAMP concentrations
### Pull-down assay of RAP1
To measure EPAC1 activity, we performed RAP1 pull-down experiments using a GST fusion protein containing the RAP1 binding domain of RAL-GDS as previously described (*Maillet et al., 2003*).

### EPAC-based BRET sensor assay (CAMYEL sensor)
EPAC1 activity was assessed using CAMYEL sensor (*Brown et al., 2014*), a BRET-based cAMP construct, composed of EPAC1 sandwiched between Renilla luciferase and citrine. NRVM were lysed, centrifuged (20,000 × $g$, 10 min), and supernatant was treated (RT, 5 min) with 2 µM coelenterazine before measure. Emission from Renilla luciferase and citrine was measured simultaneously at 465 nm and 535 nm in a plate-reader (Beckman coulter, Villepinte, France). EPAC1 activation decreases BRET signal (465 nm/535 nm ratio).

### cAMP concentration
Cardiomyocytes cAMP concentration was assessed with cAMP dynamic 2 kit (Cisbio, Saclay, France), according to the manufacturer's instructions.

## Calcium handling measurement
Cells were loaded with Fluo-3 AM as previously described (*Llach et al., 2019*). $[Ca^{2+}]_i$ transients were recorded following electrical stimulation at 2 Hz (two parallel Pt electrodes). Images were obtained with a laser scanning confocal microscope (TCS SP5X, Leica Microsystems, Nanterre, France) equipped with an ×40 water immersion objective in the line scan mode (1.43 ms/line). Fluo-3 AM was excited with a white laser fitted at 500 nm, and emission measured at wavelengths above 510 nm. Image analyses were performed by IDL 8.2 software (Exelis Visual Information Solutions, Inc) and homemade routines (*Llach et al., 2019*).

## Quantification of TopIIβ/DNA complex
The formation of TopIIβ/DNA complex was assessed by band depletion assay, as previously described (*Wartlick et al., 2013*).

## Mitochondrial respiratory chain complexes activities
NRVM were homogenized in ice-cold HEPES buffer and incubated (4°C, 1 hr) for extraction.

### Mitochondrial respiratory complex I (NADH:ubiquinone oxidoreductase) activity
Complex I activity was measured by recording the decrease in absorbance at 340 nm caused by oxidation of NADH to $NAD^+$ in phosphate buffer supplemented with decylubiquinone. In a second step, rotenone was added to compare total NADH oxidation in cells and NADH oxidation due to complex I activity.

### Mitochondrial respiratory complex IV (cytochrome *c* oxidase) activity
Cytochrome *c* oxidase activity was assayed by the decrease in absorbance at 550 nm caused by oxidation of ferrocytochrome *c* to ferricytochrome *c* by cytochrome *c* oxidase in phosphate buffer.

## Statistics
Results are expressed as mean ± SEM. Outliers were removed according to ROUT method. Differences between two groups have been analyzed by non-parametric Mann-Whitney test. The comparison between more than two groups was analyzed by Kruskal-Wallis test followed by post hoc

test with Bonferroni correction. Differences were considered significant at *p<0.05, **p<0.01, and ***p<0.001 vs Ctl and # vs Dox alone.

## Acknowledgements

We are grateful to Dr. E Hirsch and Dr. A Ghigo (University of Torino, Italy) and Dr. D Hilfiker-Kleiner (Marburg University Medical School, Germany) for providing cancer cell lines, and to V Domergue and the IPSIT platform for animal housing. Sources of funding: This works was supported by grants from Agence Nationale de la Recherche (ANR-13-BSV1-0023 and ANR-15-CE14-0005), LabEx LERMIT (ANR-10-LABX-0033), DHU TORINO, Leducq Foundation for Cardiovascular Research (19CVD02), and EU MILEAGE project #734931. AL was recipient of a Lefoulon Delalande fellowship.

## Additional information

### Funding

| Funder | Grant reference number | Author |
|---|---|---|
| Agence Nationale de la Recherche | ANR-13-BSV1-0023 | Ana-Maria Gomez |
| Agence Nationale de la Recherche | ANR-15-CE14-0005 | Jean-Pierre Benitah<br>Ana-Maria Gomez |
| LabEx LERMIT | ANR-10-LABX-0033 | Marianne Mazevet<br>Rodolphe Fischmeister<br>Eric Morel |
| DHU TORINO | | Marie-Catherine Vozenin |
| Leducq Foundation for Cardiovascular Research | 19CVD02 | Delphine Dayde<br>Rodolphe Fischmeister<br>Eric Morel |
| EU MILEAGE | project #734931 | Jean-Pierre Benitah<br>Rodolphe Fischmeister<br>Ana-Maria Gomez |
| Lefoulon Delalande fellowship | Graduate Student Fellowship | Anna Llach |

The funders had no role in study design, data collection and interpretation, or the decision to submit the work for publication.

### Author contributions

Marianne Mazevet, Maxance Ribeiro, Conceptualization, Formal analysis, Investigation, Visualization, Writing – original draft; Anissa Belhadef, Formal analysis, Investigation, Visualization, Writing – original draft; Delphine Dayde, Validation, Visualization, Writing – review and editing; Anna Llach, Marion Laudette, Tiphaine Belleville, Formal analysis, Investigation, Visualization; Philippe Mateo, Resources, Formal analysis, Supervision, Writing – review and editing; Mélanie Gressette, Formal analysis, Supervision, Investigation, Visualization; Florence Lefebvre, Supervision, Investigation; Ju Chen, Christilla Bachelot-Loza, Catherine Rucker-Martin, Frank Lezoualch, Marie-Catherine Vozenin, Resources; Bertrand Crozatier, Jean-Pierre Benitah, Writing – review and editing; Rodolphe Fischmeister, Ana-Maria Gomez, Resources, Writing – review and editing; Christophe Lemaire, Conceptualization, Supervision, Methodology, Writing – original draft; Eric Morel, Conceptualization, Supervision, Funding acquisition, Writing – original draft, Project administration, Writing – review and editing

### Author ORCIDs

Delphine Dayde http://orcid.org/0000-0001-8543-2352
Marie-Catherine Vozenin https://orcid.org/0000-0002-2109-8073
Rodolphe Fischmeister http://orcid.org/0000-0003-2086-9865
Eric Morel http://orcid.org/0000-0003-1960-0121

### Ethics

All animal experiments were conducted in line with the French/European Union Council Directives for the laboratory animals care 86/609/EEC, (MESRI 18927 authorization).

### Decision letter and Author response

Decision letter https://doi.org/10.7554/eLife.83831.sa1
Author response https://doi.org/10.7554/eLife.83831.sa2

---

## Additional files

### Supplementary files
• MDAR checklist

### Data availability

All data generated or analysed during this study are included in the manuscript and supporting file; Source Data files have been provided for Figures 1 to 7 and Figure 5-figure supplement 1.

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
