## [Editor Report]

The clinical implications of doxorubicin-induced cardiac toxicity are well established. The discovery of novel druggable cardioprotective targets has significant biological and clinical impact.

---

## [Decision Letter]

**Decision letter after peer review:**

Thank you for submitting your article "EPAC1 inhibition protects the heart from doxorubicin-induced toxicity" for consideration by *eLife*. Your article has been reviewed by 2 peer reviewers, and the evaluation has been overseen by a Reviewing Editor and a Senior Editor. The following individual involved in the review of your submission has agreed to reveal their identity: Cecilia Mundiña-Weilenmann (Reviewer #2).

The reviewers have discussed their reviews with one another, and the Reviewing Editor has drafted this letter to help you prepare a revised submission.

Essential revisions:

1) Extended Figure 2 – why do the WT and KO mice weights vary at the start … 25 vs >27.5g? This can make a difference in the way the weights changed over time as is evident in the way both Saline groups performed across the weeks.

2) Request raw uncropped blot image for Figure 2a.

3) Figure 4a: Some of the bar references are missing.

4) Figure 4i – What happens if you use a higher dose of CE3F4?

5) Figure 4i – Not 'most'. Figure disagrees with this assertion. Perhaps show a magnified image too for better clarity.

6) Figure 4k: Please specify the time of Dox-treatment in adult cardiac myocytes.

7) There is no reference to Figure 5a in the text.

8) Figure 5d: The level of EPAC1 expression in the representative blot of 15 weeks after the last i.v. injection of saline solution in WT mice is considerably higher than that of 2 and 6 weeks after the last injection of saline solution. Is there any change in the basal expression of EPAC1 with age? Request authors to send in uncropped data for Figure 5d. A full Western blot is requested.

9) Figure 5e – can authors explain why there is an apparent 4-fold difference between mice in Actin levels? As a further comment, the 2 WT mice Actin EPAC1 levels are quite different too. Was there technical difficulty in collecting protein?

10) Figure 6 a-c: The authors claimed that: "while cardiomyocytes isolated from Dox-treated WT mice had a reduced unloaded cell contraction (cell shortening, Figure 6a), a reduced [ca^2+^]i transient amplitude (peak F/F0, Figure 6b), and a decreased sarcoplasmic reticulum ca^2+^ load (Figure 6c), all these alterations were absent in myocytes isolated from Dox-treated EPAC1 KO mice". In fact, in cardiomyocytes from EPAC1 KO mice, Dox-treatment enhanced Ca transient amplitude and sarcoplasmic reticulum Ca load. Can the authors comment on that?

11) Figure 6d – unacceptable, as actin bands are heavily pixelated. Provide a more resolved image. Did the authors observe a slowdown in the kinetics of Ca transient decay as a consequence of the decrease in SERCA2a expression in myocytes from Dox-treated WT mice?

12) Prevention of Dox-induced cardiac dysfunction in WT mice by pharmacological inhibition of EPAC1, would strengthen the putative clinical application of the current findings.

13) Recommend including https://academic.oup.com/jpp/article/65/2/157/6132943 citation as it is a comprehensive review of the toxic effects of Dox, including its effects on the heart.

14) 'genetic invalidation' – change this terminology.

15) 'human cancer' – Be specific here. Which types of cancer cell lines were studied exactly?

16) 'EPAC1 inhibition was shown to inhibit cell migration and induce apoptosis' … So, are you able to show the combined effect of EPAC1i and Dox on breast cancer cells in your study? Or maybe the cervical line you were using?

17) You did not use females due to hormonal influence – provide a citation as evidence of this phenomenon.

18) Dissociation of Neonatal (NRVM) and Adult (ARVM) Rat Ventricular Myocytes – How did you ensure you got a high purity yield of your target cell type?

[Editors' note: further revisions were suggested prior to acceptance, as described below.]

Thank you for resubmitting your work entitled "EPAC1 inhibition protects the heart from doxorubicin-induced toxicity" for further consideration by *eLife*. Your revised article has been evaluated by a Senior Editor and a Reviewing Editor.

The manuscript has been improved but there are some remaining issues that need to be addressed, as outlined below:

– The version of the manuscript uploaded does not indicate which figure is which at the end where all figures are presented. On page 31, the figure 'D', can the authors explain why in the EPAC KO Dox lane, the bands for the two markers are not aligned, ie. one drops L-R, while the other rises L-R. Are they from the same experiment, as the lane to the immediate left bands look quite okay. This needs to be confirmed by having a look at the whole blot.

– Figure 6A: for the KO condition, why aren't the upper bands shown? The cropping is too close (compare with the WT condition where a wider region is shown above band of interest).

– Also Figure 6D graph y-axis should be 1-100%, and higher, not only 0-1 and higher. Compare 6E.

– Regarding the *in vivo* weight data there should be an explanation along the lines of slopes of weight curves as supplementary data (ie. Supp Figure 2A and B). A 3.5g difference is > 10% of the weight of animals.

---

## [Author Response]

Essential revisions:1) Extended Figure 2 – why do the WT and KO mice weights vary at the start … 25 vs >27.5g? This can make a difference in the way the weights changed over time as is evident in the way both Saline groups performed across the weeks.

The figure name has been modified according to *eLife* instructions as follow: Figure 5—figure supplement 1 (line 176, 760 and 766).

The animals were raised in our animal core facility at the Faculty of Pharmacy with a limited number of transgenic animals. All animals (transgenic and associated littermate) used for experimentation were 12 weeks +/- 3 days of age, which could partly explain the observed variations in weight. However, following your comment, we compared the slope of weight gain between WT and Epac1 KO Ctl mice, and between Dox treated WT and Dox Treated Epac1 KO mice. As you can see in appendix of revisions (Figure 5—figure supplement 1a/b), there was no difference between the slopes (P=0.0708 and P=0.6487, respectively).

2) Request raw uncropped blot image for Figure 2a.

We have replaced the previous blot with a new one. The uncropped raw data are provided.

3) Figure 4a: Some of the bar references are missing.

The missing bar references have been added.

4) Figure 4i – What happens if you use a higher dose of CE3F4?

The dose of CE3F4 was chosen according to the previous study performed by Courilleau et al. (Biophysical Research Communications, 2013)^1^. We used the R-CE3F4 enantiomer at 10 M based on an IC50 value of R-CE3F4 on recombinant Epac1 of 5.8 M. From our data, this concentration of CE3F4 was sufficient to produce a biological response, namely a decrease in DNA damage, cleaved caspase 3 and 9, and cell death. We preferred not to increase the dose of CE3F4 to remain selective on EPAC1 and to avoid potential side effects such as protein denaturation, e.g. as documented with ESI-09 when used at >25 M (Rehmann *et al.,* 2013^2^, Zhu *et al.,* 2015^3^).

5) Figure 4i – Not 'most'. Figure disagrees with this assertion. Perhaps show a magnified image too for better clarity.

We apologize if the old Figure 4i was not clear enough. We now provide new micrographs which we hope are more convincing. We would like to keep the description as proposed in the text (“…most CE3F4 supplemented-cells exhibited normal shape.”) because this is really what reflects the results consistently obtained in cell culture.

6) Figure 4k: Please specify the time of Dox-treatment in adult cardiac myocytes.

The time was 24h. This information has been added in the legend to Figure 4k (line 703).

7) There is no reference to Figure 5a in the text.

The reference to Figure 5a has been added in the text (line 176).

8) Figure 5d: The level of EPAC1 expression in the representative blot of 15 weeks after the last i.v. injection of saline solution in WT mice is considerably higher than that of 2 and 6 weeks after the last injection of saline solution. Is there any change in the basal expression of EPAC1 with age? Request authors to send in uncropped data for Figure 5d. A full Western blot is requested.

The original membranes have been reconstructed since they were originally cut out for separated antibodies incubation/revelation and time saving. They were again incubated with EPAC1 and actin antibodies in order to show the full membrane as requested. In addition, the “15 weeks” membrane coming from the mice after 15 weeks of treatment (saline or Dox) has been incubated/revealed with GAPDH as reference protein. These raw data are available in Figure 5—figure supplement 1.

Concerning the interesting question on the increase in basal expression of EPAC1 with age, we have no clear answer. We found one report showing that EPAC1 expression in heart is increasing during development, with a 140% increase measured at 3 weeks and 12 weeks in comparison to the foetal stage and neonatal stage (Ulucan *et al.* 2007^4^). In any case, EPAC1 expression was strongly inhibited by doxorubicin as indicated in the text.

9) Figure 5e – can authors explain why there is an apparent 4-fold difference between mice in Actin levels? As a further comment, the 2 WT mice Actin EPAC1 levels are quite different too. Was there technical difficulty in collecting protein?

These experiments have been repeated using heart tissues from the same mice and the resulting western blots are shown in new Figure 5e.

10) Figure 6 a-c: The authors claimed that: "while cardiomyocytes isolated from Dox-treated WT mice had a reduced unloaded cell contraction (cell shortening, Figure 6a), a reduced [ca^2+^]i transient amplitude (peak F/F0, Figure 6b), and a decreased sarcoplasmic reticulum ca^2+^ load (Figure 6c), all these alterations were absent in myocytes isolated from Dox-treated EPAC1 KO mice". In fact, in cardiomyocytes from EPAC1 KO mice, Dox-treatment enhanced Ca transient amplitude and sarcoplasmic reticulum Ca load. Can the authors comment on that?

A similar increase in Ca transient amplitude and sarcoplasmic reticulum Ca load was observed in our previous study (Llach et al. 2019^5^) in WT mice after 6 weeks of Dox treatment, as compared to a 2-week treatment, while both parameters were depressed again after 15 weeks of treatment (as seen in this study). This intermediate phase, which was accompanied by an increase in the expression of SERCA 2A and a decrease in the expression of phospholamban, was considered as a transient compensatory cardiac response to Dox, followed by a decompensation at 15 weeks. If, as proposed in our current study, the absence of EPAC1 might prevent the long-term depression of cardiac function induced by Dox, it may not affect the compensatory phase which in this case would last much longer that in WT mice.

We have added a small discussion in the revised manuscript (line 189-194).

11) Figure 6d – unacceptable, as actin bands are heavily pixelated. Provide a more resolved image. Did the authors observe a slowdown in the kinetics of Ca transient decay as a consequence of the decrease in SERCA2a expression in myocytes from Dox-treated WT mice?

As in the comment 8, original membranes have been reconstructed since they were originally cut out for separated antibodies incubation/revelation and time saving. They were again incubated with SERCA 2A and calsequestrin antibody in order to have a better protein reference and to show the full membrane as requested. Actin images have been replaced by calsequestrin one and the uncropped raw data are provided in the source data files.

Concerning the kinetics of Ca transient decay, we indeed observed a 20% slowdown in the kinetics of the Ca transient decay as shown in a new panel e in Figure 6 (see additions in line 195-196, Figure 6e and line 748-750). A similar observation was reported in our previous study (Llach et al. 2019^5^).

12) Prevention of Dox-induced cardiac dysfunction in WT mice by pharmacological inhibition of EPAC1, would strengthen the putative clinical application of the current findings.

This is obviously our current goal and experiments are already undertaken to test this hypothesis. Our plan is actually to use mice subcutaneously injected with tumor cells and treat the mice with saline, Dox, CE3F4 or Dox+CE3F4 (see also our response to comment 16 below). However, as shown in two recent studies, the use of CE3F4 in *in vivo* studies is challenging due to its poor plasmatic stability^6, 7^. Therefore, it will take a substantial amount of time and effort to complete these experiments. We hope the reviewers will understand that this is not achievable within the revision of this manuscript, and will rather constitute a whole and separate study.

13) Recommend including https://academic.oup.com/jpp/article/65/2/157/6132943 citation as it is a comprehensive review of the toxic effects of Dox, including its effects on the heart.

This excellent review has been added in the introduction (ref 2 in line 53 and line 456-457 for reference).

14) 'genetic invalidation' – change this terminology.

Corrected: the original terminology has been replaced by the following: “Here, we demonstrated that pharmacological inhibition (CE3F4) and the use of EPAC1 knock-out mice prevented…” (line 224).

15) 'human cancer' – Be specific here. Which types of cancer cell lines were studied exactly?

The cancer cell lines were already indicated in the Methods section. We have added the details in the Discussion (line 231): “(MCF-7 human breast cancer and HeLa human cervical cancer)”.

16) 'EPAC1 inhibition was shown to inhibit cell migration and induce apoptosis' … So, are you able to show the combined effect of EPAC1i and Dox on breast cancer cells in your study? Or maybe the cervical line you were using?

Actually, we do have preliminary experiments which shows this. As shown in Author response image 1, addition of 10 mM of CE3F4 potentiates the effect of doxorubicin on two parameters measured in HeLa cancer cell line: SubG1 cells (Author response image 1) and healing velocity (Author response image 1). However, as indicated in the response to comment 12 above, these results are part of another study comparing the effect of Dox, CE3F4 or Dox+CE3F4 in mice subcutaneously injected with tumor cells. We hope the reviewer will accept that we prefer to hold these results for this new and separate study.

**Author response image 1. sa2fig1:** 

17) You did not use females due to hormonal influence – provide a citation as evidence of this phenomenon.

A previous study from our lab showed a differential response to doxorubicin in male and female adult rats^8^. The authors showed that “After 7 weeks of doxorubicin (2 mg/kg per week), males developed major signs of cardiomyopathy with cardiac atrophy, reduced left ventricular ejection fraction and 50% mortality. In contrast, no females died and their left ventricular ejection fraction was only moderately affected.” We have added a brief statement on this in the revised manuscript (line 325-326 and reference added in line 610-611).

18) Dissociation of Neonatal (NRVM) and Adult (ARVM) Rat Ventricular Myocytes – How did you ensure you got a high purity yield of your target cell type?

First, atria were removed before the dissociation of NRVM, and after the Langendorff perfusion for ARVM isolation. Then, NRVM go through a percoll gradient, which separates in different layers the fibroblasts and the cardiomyocytes^9^, while ARVM passed by two BSA phases in order to remove the maximal amount of non cardiomyocytes cells and ensure a high purity yield^10^.

References

1. Courilleau, D., Bouyssou, P., Fischmeister, R., Lezoualc'h, F. and Blondeau, J.P. The (R)-enantiomer of CE3F4 is a preferential inhibitor of human exchange protein directly activated by cyclic AMP isoform 1 (Epac1). *Biochemical and biophysical research communications* 440, 443-448 (2013).

2. Rehmann, H. Epac-inhibitors: facts and artefacts. *Scientific reports* 3, 3032 (2013).

3. Zhu, Y. *et al.* Biochemical and pharmacological characterizations of ESI-09 based EPAC inhibitors: defining the ESI-09 "therapeutic window". *Scientific reports* 5, 9344 (2015).

4. Ulucan, C. *et al.* Developmental changes in gene expression of Epac and its upregulation in myocardial hypertrophy. *American journal of physiology. Heart and circulatory physiology* 293, H1662-1672 (2007).

5. Llach, A. *et al.* Progression of excitation-contraction coupling defects in doxorubicin cardiotoxicity. *Journal of molecular and cellular cardiology* 126, 129-139 (2019).

6. Toussaint, B., Hillaireau, H., Cailleau, C., Ambroise, Y. and Fattal, E. Stability, pharmacokinetics, and biodistribution in mice of the EPAC1 inhibitor (R)-CE3F4 entrapped in liposomes and lipid nanocapsules. *Int J Pharm* 610, 121213 (2021).

7. Toussaint, B. *et al.* Interspecies comparison of plasma metabolism and sample stabilization for quantitative bioanalyses: Application to (R)-CE3F4 in preclinical development, including metabolite identification by high-resolution mass spectrometry. *J Chromatogr B Analyt Technol Biomed Life Sci* 1183, 122943 (2021).

8. Moulin, M. *et al.* Sexual dimorphism of doxorubicin-mediated cardiotoxicity: potential role of energy metabolism remodeling. *Circ Heart Fail* 8, 98-108 (2015).

9. Wollert, K.C. *et al.* Cardiotrophin-1 activates a distinct form of cardiac muscle cell hypertrophy. Assembly of sarcomeric units in series VIA gp130/leukemia inhibitory factor receptor-dependent pathways. *The Journal of biological chemistry* 271, 9535-9545 (1996).

10. Verde, I., Vandecasteele, G., Lezoualc'h, F. and Fischmeister, R. Characterization of the cyclic nucleotide phosphodiesterase subtypes involved in the regulation of the L-type ca^2+^ current in rat ventricular myocytes. *British journal of pharmacology* 127, 65-74 (1999).

[Editors' note: further revisions were suggested prior to acceptance, as described below.]

The manuscript has been improved but there are some remaining issues that need to be addressed, as outlined below:– The version of the manuscript uploaded does not indicate which figure is which at the end where all figures are presented. On page 31, the figure 'D', can the authors explain why in the EPAC KO Dox lane, the bands for the two markers are not aligned, ie. one drops L-R, while the other rises L-R. Are they from the same experiment, as the lane to the immediate left bands look quite okay. This needs to be confirmed by having a look at the whole blot.

We added the missing figure numbers on each figure page.

The blots in Figure 6D are from the same experiment. However, as explained in our “Essential revision comment 11”, the original membranes have been reconstructed since they were originally cut out for separated antibodies incubation/revelation and time saving. They were again incubated with SERCA 2A and calsequestrin antibody in order to have a better protein reference and to show the full membrane as requested. Actin images have been replaced by calsequestrin ones and the uncropped raw data are provided in the source data files.

The non-alignment of the bands is likely due to the fact that this is the last well of the blot and the migration might have been perturbed. This is confirmed by a LI-COR staining (all protein staining) made on this blot which is now added the source data files.

– Figure 6A: for the KO condition, why aren't the upper bands shown? The cropping is too close (compare with the WT condition where a wider region is shown above band of interest).

We are sorry, but we are not sure to understand your question. The only blots with EPAC1 KO condition are in Figure 5E and 6D. In both cases the width of the regions shown are the same between WT and KO condition. In case you are referring to differences in the EPAC1 blots of Figure 5D and E, i.e. to the presence of two bands in Figure 5E and one band in Figure 5D for EPAC1, the reason is the following: the two blots were obtained using different types of gels. The one in Figure 5D was obtained with a 10% gel, and the one in Figure 5E was obtained with a gradient gel, which separates differently the proteins. EPAC1 was thus revealed as two bands with the gradient gel while the 10% gel was giving only one band. Please note that the gel in Figure 5E was repeated to reply to “Essential revision comment 9”.

– Also Figure 6D graph y-axis should be 1-100%, and higher, not only 0-1 and higher. Compare 6E.

Again, we do not understand the point. The graph in Figure 6D shows results from western blots which are presented in each figure of the manuscript as a relative level between the treated versus control, the control value being 1 (see e.g. Figure 1F, 1G, 1J, 2A, 2B, 3A, 3B, 3C, 3D, 4E, 4F, 5D). Figure 6E, which represents the kinetics of calcium transients, are presented as the other calcium data shown in Figure 6A, B and C, i.e. as percentage of control, the control value being 100%, and thus cannot be compared with western blot data.

– Regarding the in vivo weight data there should be an explanation along the lines of slopes of weight curves as supplementary data (ie. Supp Figure 2A and B). A 3.5g difference is > 10% of the weight of animals.

We added an explanation of the slopes of the weight curves in the legend to Figure 5—figure supplement 1:

“The slopes were 0.41 and 0.34 g/week in a, and 0.21 and 0.18 g/week in b for WT and EPAC1 KO mice, respectively, and were not statistically different (p=0.0708 and 0.6487, respectively in a and b). The number of mice was 5 and 6 for Sal and 7 and 5 for Dox, in WT and EPAC1 KO, respectively.”